# Paternal Biomass Smoke Exposure in Rats Produces Behavioral and Cognitive Alterations in the Offspring

**DOI:** 10.3390/toxics9010003

**Published:** 2020-12-31

**Authors:** Larisa M. Sosedova, Vera A. Vokina, Mikhail A. Novikov, Viktor S. Rukavishnikov, Elizaveta S. Andreeva, Olga M. Zhurba, Anton N. Alekseenko

**Affiliations:** FSBSI East-Siberian Institute of Medical and Ecological Research, Angarsk, Irkutsk 665827, Russia; vokina.vera@gmail.com (V.A.V.); novik-imt@mail.ru (M.A.N.); rvs_2010@mail.ru (V.S.R.); liza.2995@mail.ru (E.S.A.); labchem99@gmail.com (O.M.Z.); alexeenko85@mail.ru (A.N.A.)

**Keywords:** biomass, rats, offspring, behavior

## Abstract

Particular concern at the present stage is the health effects of wildfires’ smoke. The aim of the study was to determine the impact of paternal biomass-smoke exposure on offspring’s behavior and cognitive abilities. Male rats were exposed to biomass smoke for four hours/day, five days/week, for four weeks. Average concentration of carbon monoxide and particulate matter of 2.5 μm PM_2.5_ in the chamber during exposure were 28.7 ± 5.3 mg/m^3^ and 1.9 ± 0.5 mg/m^3^, respectively. At the same time, high concentrations of furfural and acetaldehyde were detected in the air environment of the exposure chambers. Offspring was obtained by mating of experimental males with untreated females, immediately after the end of the exposure and after 60 days (long-term period). Offspring were tested by using the Morris water maze and open field at three months of age. Male and female offspring born by mating immediately after exposure demonstrated decreased exploratory behavior, locomotor activity, and spatial navigation, as well as increased anxiety levels. Locomotor and exploratory activity in rats of both sexes from progeny obtained after long-term exposure to smoke had no statistically significant differences when compared to the control; however, the females showed a high level of anxiety and impaired cognitive functions. The recovery period after biomass-smoke intoxication, comparable in duration of spermatogenesis in rats, was an important factor in reducing the risk of developing central nervous system (CNS) disorders in offspring.

## 1. Introduction

The negative impact of wildfire smoke on human health is a unique interdisciplinary problem for the modern scientific community. The regular occurrence of large-scale forest fires is accompanied by significant smoke pollution of vast territories, which often takes the character of a natural disaster and requires the adoption of scientifically based comprehensive arrangements to justify measures to protect the population [1].

Despite the fact that wildfires have recently become global, many aspects of the toxic effects of smoke and its components on human health remain unknown. When various types of forest biomass are burned, a multicomponent mixture is released into the atmospheric air that contains particulate matter <10 μm in diameter, carbon monoxide (CO), nitrogen and sulfur oxides, aldehydes, polyaromatic hydrocarbons, volatile organic compounds, chlorinated dioxins, free radicals, etc. [2,3]. Epidemiological data today convincingly prove the negative impact of smoke from wildfires on human health. The smoke from wildfires is known to be the most dangerous for young children, people suffering from respiratory and cardiovascular diseases, and pregnant women [4,5]. A survey of pregnant women during extensive wildfires in the United States showed that chronic prenatal exposure to wildfire smoke was associated with the risk of preterm birth, low birth weight, and gestational diabetes and maternal hypertension [6,7]. A link between exposure to particulate matter (PM_2.5_) from ambient air and low birth weight among term infants is highlighted in studies by Sapkota et al. (2012), Hyder et al. (2014), Sun et al. (2016), and Li et al. (2017) [8,9,10,11]. In addition, there is information about the adverse effects of ambient fine particulate environment on the male reproductive system [12,13,14,15]. Multicomponent composition of smoke and the presence of potential repro- and genotoxicants in it determines the need for a comprehensive study of the toxic effects of smoke on the reproductive system, including distant ones, manifested in offspring. At the same time, the male reproductive potential under conditions of long-term smoke pollution and the health of the offspring of exposed males has not been sufficiently studied. Of particular importance is the problem of long-term effects of biomass smoke and accumulated chemical burden of parents for the health of future generations. To assess the disorders of the reproductive system and the CNS of offspring, the most correct is experimental modeling, which makes it possible to create smoke, as close as possible to natural conditions. Consequently, the aim of this work was to study effects of paternal biomass-smoke exposure on offspring neurobehavioral function.

## 2. Materials and Methods

### 2.1. Animals and Experimental Design

A total of 230 (10 males and 60 females of parental generation; 80 males and 80 females of offspring) outbred albino rats aged 3 months and of average weight of 180 g were used for the studies. Experimental design is presented in Figure 1. Male rats were randomly divided in two groups (*n* = 5, per group): treatment group, in which the animals were exposed to biomass smoke, and control group, in which the animals were supplied with clean air into the chamber. Male rats were exposed to biomass smoke for 4 weeks, and each rat was then mated immediately after exposure and 60 days after exposure (long-term period), with 3 untreated females. All untreated females (*n* = 60) were not exposed and were kept under standard vivarium conditions. Females were placed in separate cages two days before the expected date of birth. Offspring born by mating immediately after exposure (F1 (1)) and offspring born by mating in the long-term period after exposure (F1 (2)) were kept in standard vivarium conditions and weaned at postnatal day 30. Behavioral and cognitive indicators were assessed in adult offspring (3 months), in open field test and Morris water maze (*n* = 20 in each group). All the animals were kept under 12/12 h light/dark cycle, on a ventilated shelf, and under controlled temperature (22–25 °C) and humidity conditions (55–60%). Experimental animals were born by their own reproduction in the vivarium of Federal State Budgetary Scientific Institution “East Siberian Institute of Medical and Ecological Research” (FSBSI ESIMER) and kept on a standard diet. All animal experiments were approved by the ethical committee of FSBSI ESIMER (identification code, E32/19; date of approval, 10 September 2019, amended/approvals every 6 months) and carried out in compliance with the rules of humane treatment of animals in accordance with the requirements of the International Recommendations for Biomedical Research Using Animals (WHO, Geneva, 1985), UK Animals (Scientific Procedures) Act (UK, 1986), and National Institutes of Health Guide for the Care and Use of Laboratory Animals (NIH Publications No. 8023, revised 1978).

### 2.2. Exposure to Biomass Smoke

The rats were exposed to biomass smoke in 0.2 m^3^ chambers by whole-body inhalation during 4 weeks (5 days/week, 4 h/day). We simulated a smoke of biomass smoldering in the exposure chambers, close to full-scale in terms of the main components of the combustion parameters of forest biomass—CO and PM_2.5_. Biomass was used as a combustible substrate, which completely burns out under the conditions of a real ground fire and represents forest litter, branches, pieces of bark, and the upper soil horizon. Biomass was taken out of an ecologically clean area, away from residential areas and roads.

The experimental setup included an exposure chamber with a volume of 200 L, designed for placement of experimental animals and equipped with devices for controlling the air environment (temperature and humidity); it was also connected to a smoke-generating device, where the process of substrate smoldering took place. The amount of biomass, which was daily placed in the smoke generator, was completely burned out after 4 h of exposure, without additional access to oxygen. The smoke emitted from the smoke generator was mixed before entering the chamber with clean air, pumped by a compressor, to adjust the concentration of the gases and solids to be examined. Then the diluted smoke was directed into a pipe with uniform holes located along the upper perimeter of the exposure chamber. A fan was installed on the horizontal upper wall of the chamber, to mix the air. The animals were placed on a special mesh floor of the exposure chamber. To take air samples in the exposure chamber at the level of the animal breathing zone, a probe was installed to monitor the concentration of the gases and solid particles under study. Climatic conditions in the exposure chamber were stable during the exposure, temperature was 24–25 °C, and relative humidity was 40–60%.

Concentrations of CO, PM_2.5_, formaldehyde, furfural, and acetaldehyde chambers were determined daily, for 4 weeks, every hour after the start of the exposure. The concentration of CO was measured with a GANK-4 gas analyzer (Pribor Research and Production Association, Moscow, Russia). Mass concentration of PM_2.5_ was measured by a piezobalance dust monitor Kanomax 3521 (Kanomax Inc., Andover, NJ, USA). A general qualitative analysis of the air in the chamber and a qualitative one for volatile aldehydes were carried out, using an Agilent 5975 gas chromatography–mass spectrometer (Agilent Technologies, Santa Clara, CA, USA). The general qualitative composition of the air was carried out by taking an air sample onto a microfiber by solid-phase microextraction (SPME) for 10 min. Then, the microfiber SPME was analyzed on a gas chromatography–mass spectrometer. Separation of the components was carried out on an HP-5ms capillary column (30 m, 0.25 mm, and 0.25 μm), in a temperature-gradient mode. The scanning range of masses was 30–500 amu (atomic mass unit). Components were identified by using the NIST mass spectra library. Qualitative analysis for volatile aldehydes was carried out by using the reagent o-pentafluorobenzylhydroxylamine (PFBHA). For this, distilled water was taken from the air. The selected aldehydes in distilled water were derivatized with the o-PFBHA reagent; the derivatized aldehydes were extracted with hexane. The hexane extract was analyzed on a gas chromatography–mass spectrometer. The mass scanning range was 35–500 amu. Components were identified by using the NIST mass spectra library.

### 2.3. Open-Field Test

Exploratory behavior, anxiety-like behavior, and general locomotor activity were examined, using the open-field test. Each rat was placed in the center of the apparatus consisting of a round area (97 cm in diameter) surrounded by white acrylic walls (42 cm high). Within 3 min, horizontal (number of squares crossed) and vertical motor activity (rearing), holes explored (sniffing at a hole or actively moving a lid), defecation, and episodes of grooming and freezing (fading) were recorded. The test chamber was illuminated at 100 lux.

### 2.4. Morris Water Maze

The Morris water maze was a circular pool that was 1.5 m in diameter and 60 cm high, filled with water, at a temperature of about 25 °C, to a height of 25 cm, clouded by the addition of chalk. The top surface of the hidden platform was 14 cm in diameter and 1.5 cm below the surface of the water. Animals were tested four times (with an interval of 60 s), sequentially, from different sectors of the pool, while the location of the platform hidden under the water remained constant. If the animal did not find the platform within 60 s, it was forcibly placed on it. The time spent on the platform was 60 s. The latency to find the hidden platform was recorded.

### 2.5. Statistical Analyses

Statistical analysis of the research results was carried out by using the Statistica 6.1 software package. The Shapiro–Wilk W-test was used to decide on the type of feature distribution. To compare groups, we used the Mann–Whitney U-test and the two-sided Fisher test. Null hypotheses about the absence of differences between the groups were rejected at the achieved significance level of *p* ≤ 0.05.

## 3. Results

### 3.1. Exposure Characteristics

In experimental modeling of wildfire concentration of CO, PM_2.5_, furfural, and acetaldehyde in the exposure chamber were higher than WHO guideline (Table 1). The highest CO concentrations in the chamber were determined 2–3 h after the start of the seed; during this period, the CO content was 30.5–61.4 mg/m^3^, which, probably, can be associated with a more intense smoldering process, due to a decrease in the initial substrate moisture.

The results of qualitative analysis of the total composition of the air have shown that the exposure chamber essentially contains compounds of the following classes: the heterocyclic aldehydes, terpenes, terpene derivatives, aromatic hydrocarbons, and phenols (Figure 2).

Results of qualitative analysis for volatile aldehydes showed that the exposure chamber essentially contains the following types of carbonyl compounds: aldehydes, linear C 1-C 6 ketones–acetone, diketones–dimetilglioksal, and heterocyclic aldehydes–2-furaldehyde. There was a significant selection of the following compounds: acetaldehyde, formaldehyde, propionaldehyde, and dimetilglioksal (diacetyl) (Figure 3).

### 3.2. Behavioral Data

#### 3.2.1. Open Field Activity

Locomotor activity of both males and females offspring born by mating immediately after exposure was suppressed, as evidenced by a statistically significant increase in the duration of immobility, in comparison with the control groups (*U* = 6, *Z* = −3,14, *p* = 0.002 and *U* = 5, *Z* = −3,20, *p* = 0.001, respectively, Table 2). At the same time, the females showed a deficiency of exploratory behavior, characterized by a decrease in the duration of “sniffing” and the number of holes explored (*U* = 8, *Z* = 2.98, *p* = 0.002 and *U* = 19, *Z* = 2.10, *p* = 0.037, respectively, Table 2). In addition, individuals of both sexes showed a high level of anxiety, determined by the number of freezing episodes, which was six to seven times higher than the indicators of the corresponding control groups (in both cases: *p* < 0.001; Mann-Whitney U-test, Table 2). 

#### 3.2.2. Morris Water

Results from the Morris water-maze testing are summarized in Figure 4, Figure 5 and Figure 6. Males offspring born by mating immediately after exposure showed a statistically significant increase in the latency period for the search for a hidden platform (*U* = 16, *Z* = −2.08, *p* = 0.03, Figure 4), which may indicate a decrease in spatial navigation and memory.

In offspring born by mating in the long-term period of intoxication with biomass smoke, there were no violations of locomotor and exploratory activity revealed in the open field. The latent period of the search for a hidden platform in male and female white rats did not have statistically significant changes when compared with the corresponding indicators of the control groups (Figure 5). However, a statistically significant increase in the number of freezing acts in the group of females from the offspring obtained (*U* = 67, *Z* = −2.04, *p* = 0.041, Table 2), which is an indicator of a high level of anxiety, is noteworthy. In addition, when tested in the Morris water maze, among these individuals, 35% did not cope with the test, i.e., they could not find a platform in four attempts (*p* = 0.01; Fisher’s test, Figure 6). Other groups of animals demonstrated performance of the test in the 82–100%.

## 4. Discussion

This study shows that paternal biomass exposure, prior to breeding, resulted in offspring that demonstrated significantly more anxiety and reduced locomotor and exploratory activity, as well as impaired spatial memory. It should be noted that the most pronounced changes in behavior were observed in female offspring. There is evidence of sex differences in the level of stress reactivity in the offspring of fathers exposed to stress. The results of this study are consistent with those of Saavedra-Rodriguez and Feig, which showed that chronic social stress in male mice experienced during adolescence and early adulthood can cause social deficits and increased anxiety behavior in female offspring [16]. In addition, studies by Morgan and Bale showed that epigenetic changes among stress regulator genes, primarily glucocorticoid receptor genes, are observed in offspring of fathers under stress, and that these effects are likely to be sex dependent [17]. Thus, exposure of male rats to the chronic stress of forced swimming leads to an increase in the level of DNA methylation of the glucocorticoid receptor gene (Nr3c1) in the hippocampus of the offspring [18].

At the next stage, mating of exposed males with intact females was carried out 60 days after exposure, which corresponds to the length of a full spermatogenic cycle in rats. Offspring born by mating in a long-term period after exposure to biomass smoke showed normalized behavioral parameters, especially in the group of males. In female offspring, the active search component of behavior restored to background values; however, the level of anxiety in these rats was significantly increased. Perhaps the high level of anxiety caused them to fail to perform the task of spatial navigation in the Morris water maze. The results obtained are consistent with the data of Short et al., which indicate that prolonged stress in fathers induces changes in the male reproductive system and leads to dysregulation of the stress response in offspring [19].

A growing body of experimental research now confirms that the health and epigenetic profile of offspring depends on preconception paternal exposures, such as exposure to tobacco smoke [20], ethanol, opiates, psychostimulants [21], endocrine disruptors [22], and ionizing radiation [23], as well as nutritional status [24,25]. We can hypothesize that behavioral dysfunction in offspring is associated with alterations in sperm DNA methylation. The high oxidative potential of PM in forest-fire smoke [26] and the presence of potential gaseous genotoxicants in it can cause pathological conditions leading to DNA damage and fragmentation and apoptosis of spermatozoa. According to numerous studies, air-pollution ultrafine particles and gases affects the quality of sperm and the level of DNA fragmentation in sperm [27,28,29], which confirms and makes convincing our hypothesis of probable changes in the genome and/or epigenome of male germ cells, leading to behavioral and cognitive impairments in offspring born by mating males immediately after exposure. To date, there are no known published studies examining the offspring of wildfire-smoke-exposed male rats, although there are a few with maternal exposure in the prenatal period. Gorbatova et al. reported that exposure of pregnant rats to peat smoke causes DNA damage in placenta and embryos, inhibited body weight gain in the progeny, and impaired the loss of orientation and exploratory behavior during repeated testing [30,31]. It should be noted that this methodological approach does not exclude direct smoke exposure to the pups in utero.

## 5. Conclusions

The analysis of the results of the experimental study showed that the impact of a multicomponent mixture containing PM and organic compounds on the parental generation causes a change in the functional state of the central nervous system (CNS) in offspring, assessed by ethological indicators of the structure of behavior. It is characterized by a significant decrease in motor and exploratory activity against the background of increased level of anxiety, along with disturbances in spatial memory indices. The recovery period after prolonged biomass-smoke intoxication from wildfires is important to reduce the risk of developing CNS disorders in offspring. The duration of the long-term period in the experiment was 60 days, which is comparable to the timing of spermatogenesis in rats, and was sufficient for significant, although not complete, prevention of negative effects of smoke exposure in the offspring of exposed males. Confirmation of the results of experimental modeling on animals obtained by us requires clinical observation on a large sample of people. Only in this case, the possible risks to the health of children born to parents who have been in the center of smoke from the biomass of forest fires can be identified.

## Figures and Tables

**Figure 1 toxics-09-00003-f001:**
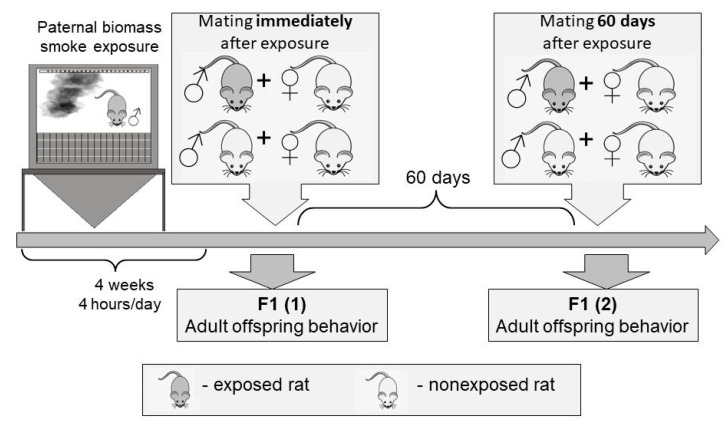
Design of the experimental study. Experimental design: Male rats were exposed to biomass smoke (4 hours/day, 5 days/week, for 4 weeks). Exposed or control males were mated with unexposed female rats, to generate the offspring: immediately after exposure and after 60 days (long-term period). Behavior and cognitive abilities of offspring born by mating immediately after exposure (F1 (1)) and offspring born by mating in the long-term period after exposure (F1 (2)) were assessed.

**Figure 2 toxics-09-00003-f002:**
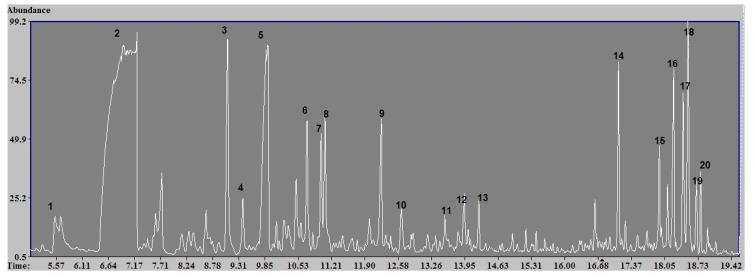
Chromatogram of the components selected by the SPME method. Identified components: (1) Hexanal, (2) furfural, (3) α-Pinene, (4) Camphene, (5) methylfuraldehyde, (6) 3-carene, (7) *p*-cymene, (8) D-limonene, (9) o-Guaiacol, (10) levoglucosenone, (11) Borneol, (12) Creosol, (13) Berbenone, (14) Longifoline, (15) γ-Muurolene, (16) α-Muurolene, (17) γ-Cadinene, (18) δ-Cadinene, (19) α-Cadinene, and (20) α-Calacorene.

**Figure 3 toxics-09-00003-f003:**
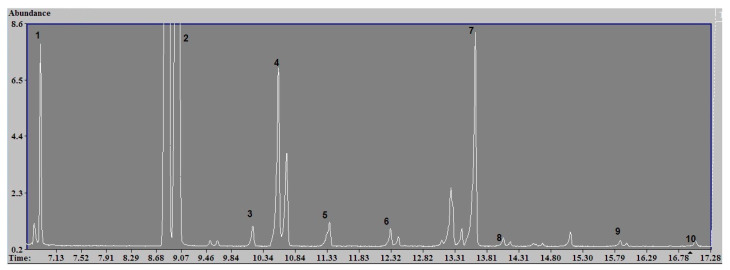
Chromatogram of derivatized aldehydes PFBHA. Identified components: (1) formaldehyde, (2) acetaldehyde, (3) acetone, (4) propionic aldehyde, (5) isobutyl aldehyde, (6) butyl aldehyde, (7) diacetyl, (8) valerian aldehyde, (9) hexanal, and (10) furfural.

**Figure 4 toxics-09-00003-f004:**
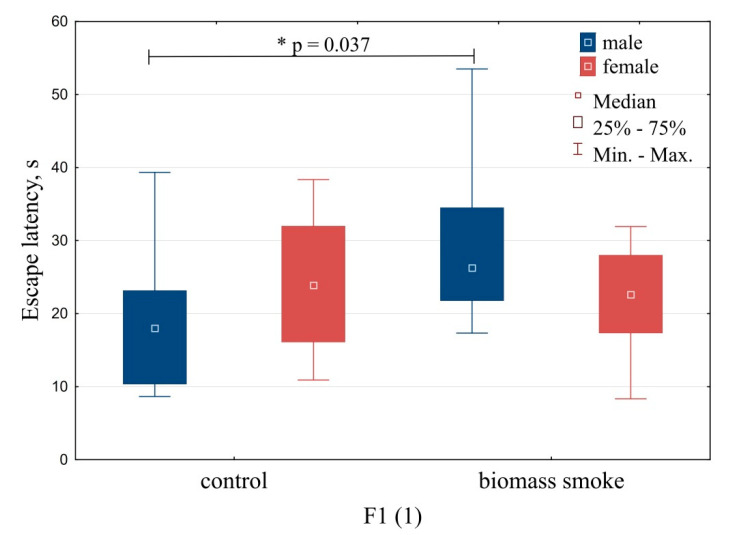
Test results in the Morris water maze. Offspring born by mating immediately after exposure. Note: * the differences statistically significant compared to the control at *p* < 0.05.

**Figure 5 toxics-09-00003-f005:**
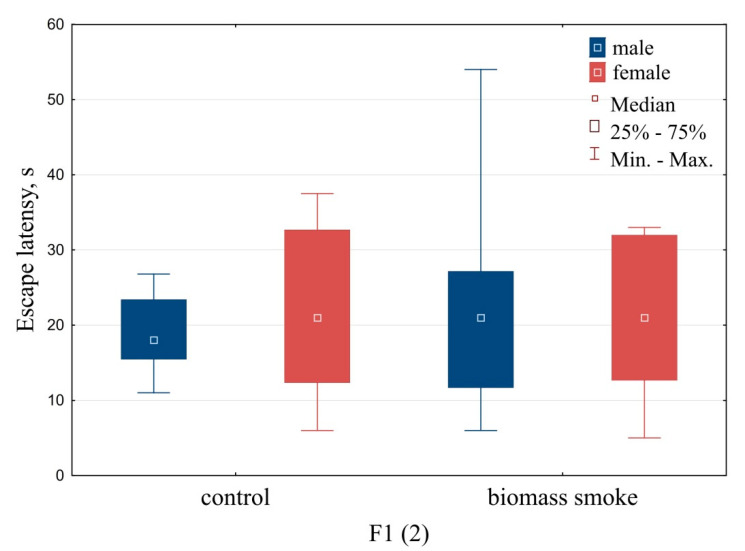
Test results in the Morris water maze. Offspring born by mating in the long-term period after exposure.

**Figure 6 toxics-09-00003-f006:**
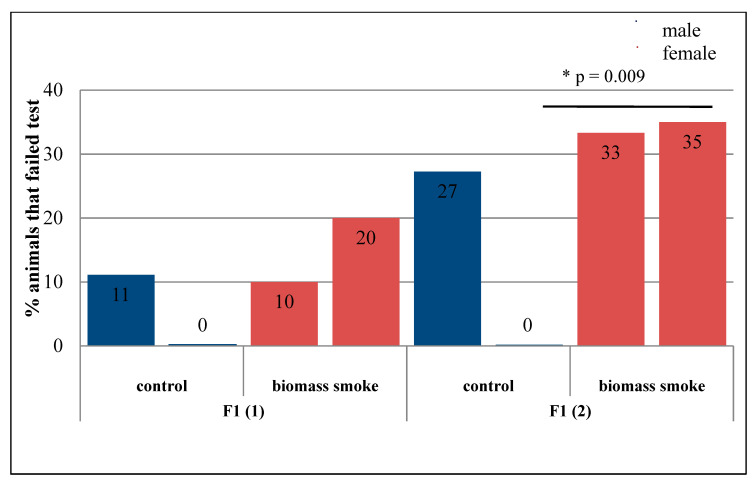
Test results in the Morris water maze. Number of animals that completed and failed the test. Note: * the differences statistically significant compared to the control at *p* < 0.05.

**Table 1 toxics-09-00003-t001:** Average concentration of PM_2.5_ and select gaseous pollutants in exposure chamber.

Contaminants	Unit	Mean ± m	Range
CO	mg/m^3^	28.78 ± 5.34	3.90–61.41
PM_2.5_	mg/m^3^	1.97 ± 0.55	0.48–4.77
Furfural	mg/m^3^	0.19 ± 0.06	0.06–0.87
Formaldehyde	mg/m^3^	0.024 ± 0.006	0.005–0.101
Acetaldehyde	mg/m^3^	0.65 ± 0.11	0.07–0.78

**Table 2 toxics-09-00003-t002:** Open field behavior in offspring of rats exposed to biomass smoke, Me (LQ;UQ).

Indicators	F1 (1)		F1 (2)	
	Males	Females	Males	Females
	*n* = 20	*n* = 20	*n* = 20	*n* = 20
Rearing	1(1;3)	0(0;2)	2.5(1;5)	3(1,5;4)
	2(0;2)	0(0;1)	3(2;4)	4.5(1;9)
Number of grooming	0.5(0;2)	0.5(0;1) *	1(0;2)	0(0;2)
	0(0;1)	0(0;0)	0(0;1)	1(0;2)
Number of freezing episodes	6(5;7) **	7(4;9) **	2(1;2)	2(0;3)*
	1(0;3)	1(1;1)	1(1;4)	1(0;2)
Holes explored	1.5(1;2)	2.5(2;4) *	1.5(1;3)	1(0;2)
	3(0;4)	5(5;7)	1(0;3)	1(0;1)
Number of defecation	0(0;0)	0(0;0)	0(0;3)	1(0;1)
	1(0;1)	0(0;1)	0(0;3)	0(0;1)
Number of squares crossed	54(51;57)	66(55;74)	41(21;59)	36(28;62)
	38(24;51)	60(46;74)	46(26;56)	33(28;48)
Sniffing, s	64(57;71)	58(42;63) *	64(52;73)	56(52;67)
	69(67;80)	77(71;80)	67(62;80)	71(66;84)
Locomotion, s	83(72;88)	88(82;91)	99(82;109)	112(98;121)
	81(64;96)	93(89;96)	96(93;101)	103(79;111)
Immobility, s	31(26;46) **	33(22;45) *	21(10;28)	20(9;36)
	16(7;18)	8(3;17)	16(12;36)	21(10;43)

Note: Me (LQ;UQ) is median and inter-quartile ranges; under the line—indicators of the corresponding control group; * the differences statistically significant compared to the control at *p* < 0.05, ** at *p* ≤ 0.01.

## Data Availability

The data presented in this study are available on request from the corresponding author.

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
