# Peer review of "Paternal Biomass Smoke Exposure in Rats Produces Behavioral and Cognitive Alterations in the Offspring"

_toxics, 2020, doi:10.3390/toxics9010003_

Round 1

Reviewer 1 Report

The article “Paternal wildfire smoke exposure in rats produces behavioral and cognitive alterations in the offspring” focuses attention of wildfire effects on male rats parental generation, demonstrating that the administration of CO and PM2.5 in an adequate chamber, lead to disturbances on offspring behavior and its cognitive abilities, and the recovery period of about 60 days after prolonged wildfires is important to reduce CNS disorders in offspring.

Till now, there are no known studies in literature about male rats, but only a few on female pregnant rats, so this work constitutes a further point of view on wildfire smoke.

With these premises, a positive judgment is expressed.

Requested changes:

In line 11, replace “is” with “was”;

Line 14: substitute “m3” with “m3

Please, the authors have to be clearer from Line 20 to line 22 in the piece “The behavior of rats of both sexes from offspring produced in the long-20 term period after smoke exposure characterized by the normalization of motor and exploratory 21 activity to background values

In line 24, the authors have to indicate what CNS stands for.

Author Response

Dear Reviewer! Authors team would like to thank you for your detailed review of the manuscript. We have tried to answer your questions as fully as possible and, if necessary, make changes to the text of the manuscript. More detailed information is presented in attached file

Reviewer 2 Report

See attached.

Author Response

(The authors gave the same response as above.)

Reviewer 3 Report

Sosedova et al describe the effect of wildfire exposure on offspring through the paternal line. They investigate the effect of the exposure on behavioral outcomes including the open field and Morris water maze test. Strength of the study is the sample size and it is generally well written. A major concern is that the control group results are far from stable. In the short term group the female offspring is more active than the male group, while in the long term group it’s mixed (higher immobility in females, but also higher locomotion)

  • How does the exposure relate to exposure in humans? It seems extreme to exposure animals 4 weeks, 4 hours a day to direct wildfire.
  • Why are analyses stratified by sex? The authors should also show the effects for all animals. Also because the results are not consistent for one sex? Also between males and females there is a large difference already in the control group between males and females? For example, immobility is 16.4 in males, 8.8 in females, while in the long term groups 15.9 and 21.4. You cannot assess the effect of an exposure if the control groups are not stable and already show an effect in itself.
  • In the discussion the authors relate to epigenetic mechanisms. I would refrain from that, because the effect could just be mutagenic instead of more regulated epigenetic mechanisms.

Minor:

  • Commas are used to indicate decimals instead of dots.

Author Response

(The authors gave the same response as above.)

Round 2

Reviewer 3 Report

-

Author Response

Dear reviewer! Thank you for reanalyzing our manuscript.